# Resolving the Duplicate-Feature Paradox with ReSHAP: A Redundancy-Weighted Generalization of Shapley Attribution

## Abstract

Shapley-value–based feature attribution methods are widely used to explain machine learning model predictions. However, these methods suffer from a critical flaw, often observed when features are duplicated, its total contribution to the model prediction is unfairly inflated, diminishing the attribution of other important features. This paradox arises because traditional Shapley-based methods allocate joint contributions equally across all participating features, regardless of redundancy or informational overlap.

In this work, we propose ReSHAP, a redundancy-aware generalization of Shapley attribution that systematically resolves the duplicate-feature paradox. ReSHAP adjusts the allocation of credit within feature coalitions by down-weighting features that contribute redundant information. We begin by proving that no attribution method can simultaneously satisfy equal division and duplication-invariance, even in instances without redundant features. This reveals a fundamental trade-off in designing fair attribution methods. Building on this insight, ReSHAP redefines how Shapley values are computed by redistributing interaction terms across feature subsets using a recursive weighting scheme, using only the standard value function without additional distributional assumptions. We support our theoretical findings with illustrative examples and experiments, highlighting the practical effectiveness of ReSHAP.

## 1 Introduction

The indisputable and growing impact of artificial intelligence (AI) on various areas of human life comes hand in hand with growing concerns about the lack of understanding of how these methods make decisions. Although the algorithms used to train such models are well understood, the outcomes of the training process often remain opaque. The lack of transparency and control over the inner workings of AI systems, combined with the massive complexity of models that often exceed human cognitive capacity, raises significant concerns about the explainability of their results. This, in turn, has sparked growing interest in Explainable AI (XAI) techniques, which aim to clarify and interpret AI model outputs Dazeley et al. (2021); Adadi & Berrada (2018); Linardatos et al. (2021); Gunning et al. (2019); Saeed & Omlin (2023). In fact, a wide range of XAI methods have emerged, reflecting the multi-faceted nature of the explainability problem. One of the most popular and widely used approaches applies the game-theoretic concept of Shapley values Shapley (1953) to the field of XAI. Shapley values originated in cooperative game theory and have been adapted to attribute an AI model's prediction to its input features in a principled way.

Shapley values offer significant explanatory power for interpreting model predictions. The method has a solid theoretical foundation that ensures a fair allocation of importance to features. These properties have made Shapley-value-based explanations highly attractive. However, the applicability of exact Shapley values remains limited due to the computational overhead of evaluating all $2^n$ feature subsets. To address this, various methods have been introduced to overcome the computational challenges. Some approaches approximate Shapley values, such as SHAP (SHapley Additive exPlanations) and Kernel SHAP, introduced in Lundberg & Lee (2017), which use a linear approximation to estimate the model output based on subsets of input features. Other methods exploit structural properties of specific model classes to compute Shapley values exactly and efficiently,

such as TreeSHAP Lundberg et al. (2020), which is tailored for tree-based models. For neural networks, DeepSHAP Lundberg et al. (2020) leverages the connection between SHAP and DeepLIFT to efficiently approximate Shapley values by backpropagating contribution scores.

Despite the explanatory power offered by Shapley values and their widespread adoption in the machine learning community, several limitations of the framework are well documented. The work Kumar et al. (2020) presents both mathematical and human-centric issues associated with the method. Some concerns, such as the fact that Shapley values only apply to decompose the difference between the model prediction and its expected value, arise from how the method is applied. These can be addressed, for example, by comparing the model output to an alternative reference value Merrick & Taly (2020), or by decomposing other types of values; see, e.g., Owen & Prieur (2017). However, other limitations, such as the *duplicate-feature paradox*, remain valid criticisms of Shapley value–based methods. The duplicate-feature paradox concerns a situation in which, for a model function $f(x_1, \ldots, x_n)$ with $n$ features, a proxy function $f'(x_1, x_1, \ldots, x_n)$ is constructed by duplicating an input feature (e.g., $x_1$), resulting in a model with $n + 1$ features. In such a case, the contribution of the remaining features $\{x_j : j \in [n] \setminus \{1\}\}$ can be significantly diminished. This is not merely a contrived pathological example; similar effects arise when features are highly correlated, for instance, when one feature is a statistical proxy of another. This simple case, which we adopt as a running example in this paper, illustrates a broader issue: subsets of features can interact with other subsets in more intricate ways, leading to distorted attributions of individual feature contributions as computed by Shapley values.

Various methods have been proposed to address the duplicate-feature paradox. Aas et al. (2021) modify the background distribution in Kernel SHAP to better approximate Shapley values under Gaussian assumptions, but this doesn't resolve the core issue in the Shapley framework. Similar background-modulating approaches, like Merrick & Taly (2020), and modeling-intensive methods such as Frye et al. (2020), reweight permutations to improve attribution, yet lack a universal principle for selecting weights. Kwon & Zou (2022) propose learning these weights from data. Basu & Maji (2022) take a different approach, decorrelating features via linear projections before computing Shapley values. Meanwhile, Owen (1977) propose a group-based method, computing Shapley values first across feature groups and then within them, effective for exact duplicates, but not subtle or cross-group dependencies. Finally, KL-divergence-based methods, such as Watson et al. (2023) and Ay et al. (2020), redefine the value function underlying Shapley attributions, offering alternatives in contexts like precedence constraints or information decomposition, posing a computational challenge to get additional information about the probability distributions.

**Contribution.** We propose a novel framework to resolve the duplicate-feature paradox. First, we prove that no attribution method can satisfy both the equal division property and resolve the paradox, even when no redundant features are present (see Theorem 9). This result is of independent interest to cooperative game theory. Building on our first result, we propose an intuitive and efficient method that resolves the duplicate-feature paradox at both individual and subset levels (see Theorem 11). It adjusts Shapley values by accounting for feature redundancy, with small computational overhead. We supplement this with examples demonstrating how redundant features distort standard Shapley values and how the ReSHAP method resolves these cases. Finally, using the real-world AMES dataset, we illustrate that for an MLP model, adding duplicated features leaves ReSHAP attributions of non-duplicated features largely unchanged, while standard Shapley values show significant shifts in their relative importance.

## 2 PRELIMINARIES

We start by defining the Shapley values more formally. For $n \in \mathbb{N}$, let $\mathcal{X} \subseteq \mathbb{R}^n$ and $\mathcal{Y} \subseteq \mathbb{R}$. Let $\Omega$ be a sample space, define a family of random variables $X_i : \Omega \mapsto \mathcal{R}$ for $i \in [n]$ and a random vector $X : \Omega \mapsto \mathcal{X}$, given by $X = (X_1, \ldots, X_n)$. We consider a model function $f : \mathcal{X} \mapsto \mathcal{Y}$, such that $f(X) : \Omega \mapsto \mathcal{Y}$ is a real-valued random variable obtained by composing $f$ with $X$.

Let $(x_1, \ldots, x_n)$ be a realization of the random vector $X$. For any subset $S \subseteq [n]$, let $\bar{S} := [n] \setminus S$. We assume that for every $S \subseteq [n]$, the conditional distribution of $X_{\bar{S}}$ given $X_S$ is well defined. Thus, we can define:

$$\nu(S) := \mathbb{E}\big[f(x_S, X_{\bar{S}}) \,\big|\, X_S = x_S\big]. \tag{1}$$

Note that the function $\nu$ should, in principle, also be parameterized by $f$ and $x$, as it depends on the model and the input point. However, for the sake of readability, we omit these extra parameters, since throughout the paper we will compute values for a fixed model $f$ and sample point $x$. In case parameters $f$ and $x$ are needed they appear in superscript, e.g., $\nu^{f,x}(S)$. A similar convention is applied to the Shapley values, defined as follows:

**Definition 1** (Shapley values). *For $i \in [n]$, the marginal contribution of feature $i$, called the* Shapley value*, is defined as:*

$$\psi_i := \sum_{S \subseteq [n] \setminus \{i\}} \frac{|S|!(n - |S| - 1)!}{n!} \left(\nu(S \cup \{i\}) - \nu(S)\right).$$

## 3 REFORMULATION OF SHAPLEY VALUES

Let $\mu : 2^{[n]} \mapsto \mathbb{R}$ be a signed set function for $T \subseteq [n]$ defined via the Möbius inversion of the value function $\nu$:

$$\mu(T) := \sum_{[n] \setminus T \subseteq S \subseteq [n]} (-1)^{|T| - |[n] \setminus S| + 1} \left(\nu(S) - \nu(\varnothing)\right). \tag{2}$$

The value $\mu(S)$, for $S \subseteq [n]$, represents the portion of the total contribution that arises uniquely from the joint interaction among the features in $S$, excluding contributions from any of the features outside $S$. In other words, it captures the pure interaction effect attributable to the combination of features in $S$. The signed measure $\mu$ can be seen as obtained by lifting $\nu$ via the Radon–Nikodym derivative. Note that the values $\mu(S)$ for $S \subseteq [n]$ form a basis that is distinct from the one derived via unanimity games, also known as Harsanyi dividends van den Brink & Funaki (2025). While the Harsanyi basis is more commonly used in the context of Shapley values, the basis induced by the $\mu$ values proves to be more convenient for the types of derivations and decompositions we aim to perform in this work.

Alternatively, one can think of $\mu$ as a partition of the space into disjoint regions such that for every $S \in [n]$ we have $\nu(S) - \nu(\varnothing) = \sum_{T \cap S \neq \varnothing} \mu(T)$. Lemma 2 proves this more formally.

**Lemma 2.** *For every $S \subseteq [n]$*

$$\nu(S) - \nu(\varnothing) = \sum_{T \cap S \neq \varnothing} \mu(T),$$

*if and only if for every $T \subseteq [n]$, $T \neq \varnothing$*

$$\mu(T) := \sum_{[n] \setminus T \subseteq S \subseteq [n]} (-1)^{|T| - |[n] \setminus S| + 1} \left(\nu(S) - \nu(\varnothing)\right).$$

*Proof.* We start by proving the 'only if' direction. It holds:

$$\nu(S) - \nu(\varnothing) = \sum_{T \cap S \neq \varnothing} \mu(T) = \sum_{T \subseteq [n]} \mu(T) - \sum_{T \subseteq [n] \setminus S} \mu(T).$$

Let $M$ be equal to $\sum_{T \subseteq [n]} \mu(T)$. For the change of variables $A = [n] \setminus S$, let $F(A) = \sum_{T \subseteq A} \mu(T)$ which gives

$$\nu([n] \setminus A) - \nu(\varnothing) = M - \sum_{T \subseteq A} \mu(T) = M - F(A). \tag{3}$$

We can use a general inclusion-exclusion formula based on Möbius inversion and zeta transformation for the function $F(A)$ Graham et al. (1996) which yields that

$$F(A) = \sum_{T \subseteq A} \mu(T),$$

if and only if

$$\mu(T) = \sum_{A \subseteq T} (-1)^{|T| - |A|} F(A). \tag{4}$$

Transforming Equation 3 for $F(A)$ and plugging into Equation 4 yields

$$\mu(T) = \sum_{A \subseteq T} (-1)^{|T|-|A|} \left( M - \nu([n] \setminus A) + \nu(\varnothing) \right)$$

$$= M \sum_{A \subseteq T} (-1)^{|T|-|A|} - \sum_{A \subseteq T} (-1)^{|T|-|A|} \left( \nu([n] \setminus A) - \nu(\varnothing) \right)$$

$$= \sum_{A \subseteq T} (-1)^{|T|-|A|+1} \left( \nu([n] \setminus A) - \nu(\varnothing) \right),$$

where the last equality holds for $T \neq \varnothing$ because of the identity $\sum_{A \subseteq T}(-1)^{|T|-|A|} = (1-1)^{|T|} = 0$. Changing the variables back to $S = [n] \setminus A$ yields

$$\mu(T) = \sum_{[n]\setminus S \subseteq T} (-1)^{|T|-|[n]\setminus S|+1} \left( \nu(S) - \nu(\varnothing) \right) = \sum_{[n]\setminus T \subseteq S \subseteq [n]} (-1)^{|T|-|[n]\setminus S|+1} \left( \nu(S) - \nu(\varnothing) \right),$$

which finishes the only if implication. To prove the 'if' direction, it suffices to reverse the steps.  □

As a result, we can redefine the Shapley values through the measure $\mu$.

**Lemma 3.** *For $i \in [n]$, the Shapley value of feature $i$ can be expressed in the form*

$$\psi_i := \sum_{T \supseteq \{i\}} \frac{1}{|T|} \mu(T).$$

*Proof.* Proof in Appendix A.

**Observation 4** (Equal division). *The alternative formulation of Shapley values given in Lemma 3 gives rise to the equal division property van den Brink & Funaki (2025), which can also be derived from the four Shapley Axioms. Although this property is classically stated with respect to the unanimity game (Harsanyi) basis, it also holds for our basis defined by the values $\mu(S)$ for $S \subseteq [n]$. Specifically, the equal division property asserts that each atomic contribution $\mu(S)$ is equally divided among all features $i \in S$.*

The equal division property is visualized in Figure 1.

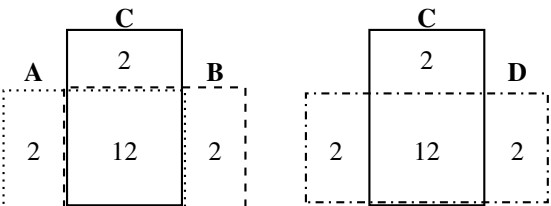

Figure 1: Venn diagrams showing $\mu$ values for examples of two functions, with 3 features on the left and 2 features on the right. The equal division property implies that in the left case $\mu(\{A, B, C\}) = 12$ is split equally among 3 features, $A$, $B$ and $C$, while in the right case $\mu(\{C, D\}) = 12$ is split equally among 2 features, $C$ and $D$.

## 4 DUPLICATE-FEATURE PARADOX

Building upon the Shapley values reformulation introduced in the *Reformulation of Shapley values* section, we more formally introduce the duplicate-feature paradox in this section and present concrete examples that highlight its impact on standard Shapley values. The *duplicate-feature paradox* refers to the phenomenon where duplicating an input feature leads to a decrease in the Shapley value of the unduplicated feature(s), while the total attribution to the duplicated features increases. This contradicts the intuitive expectation that duplicating identical information should not affect the resulting attribution. The paradox extends to the case when, instead of simply duplicating features, we add redundant features. We start with the definition of a redundant feature.

**Definition 5** (Redundant feature). *Let $f : \mathbb{R}^n \to \mathbb{R}$ be a model function, and let $\nu$ be a corresponding value function (e.g., as defined in Equation 1). A feature $j \in [n]$ is called* redundant *if there exists a subset $S \subseteq [n] \setminus \{j\}$ such that*

$$\nu(S \cup \{j\}) = \nu(S).$$

**Lemma 6.** *Let $f : \mathbb{R}^n \to \mathbb{R}$ be a model function, and let $\nu$ be a corresponding value function (e.g., as defined in Equation 1). Define a new model function $f' : \mathbb{R}^{n+1} \to \mathbb{R}$ by*

$$f'(x_1, \ldots, x_n, x_{n+1}) := f(x_1, \ldots, x_n) \quad \text{for all } x \in \mathbb{R}^{n+1}.$$

*Then, $f'$ satisfies this identity if and only if there exists a set $S \subseteq [n]$ such that*

$$\nu^{f'}(S \cup \{n+1\}) = \nu^f(S).$$

*Proof.* We prove the only if direction. Assume that for every $x \in \mathbb{R}^{n+1}$

$$f'(x_1, \ldots, x_n, x_{n+1}) := f(x_1, \ldots, x_n) \quad \text{for all } x \in \mathbb{R}^{n+1}.$$

Since by definition of value function $\nu$, Equation 1, it holds that

$$\nu^{f,x}([n]) = f(x_1, \ldots, x_n) \text{ and}$$
$$\nu^{f',x}([n+1]) = f'(x_1, \ldots, x_n, x_{n+1}),$$

the claim holds for $S = [n]$. To prove the if direction, it is enough to reverse the steps.

$\square$

**Definition 7** (Duplication-invariance). *Let $f : \mathbb{R}^n \to \mathbb{R}$ be a model function, let $f' : \mathbb{R}^{n+1} \to \mathbb{R}$ be any extended model function in which the feature $x_{n+1}$ is redundant. Let $\nu : 2^{[n+1]} \to \mathbb{R}$ be a value function (e.g., as defined in Equation 1). A feature attribution function $\phi_i \in \mathbb{R}$ for $i \in [n+1]$, computed from $\nu$, is said to be* duplication-invariant *if the following holds:*

$$\phi_i^{f'} = \phi_i^f \quad \text{for all } i \in [n] \setminus S,$$
$$\sum_{k \in S \cup \{n+1\}} \phi_k^{f'} = \sum_{k \in S} \phi_k^f,$$

*where $S \subseteq [n]$ is a minimal set satisfying the condition in Definition 5.*

The cornerstone of this research is the fact that Shapley values are not *duplication-invariant*, as we demonstrate with the following examples.

**Example 8.** *Let $f : \mathbb{R}^2 \to \mathbb{R}$ be a model function that estimates the value of real estate based on a random vector with two components: the property size $X_S$ and property location $X_L$ (both understood as real numbers). Let $x_S$ and $x_L$ be the specific values of size and location for which the model predicts the price.*

*Assume that the conditional expectations have been computed, yielding the following $\nu$ values:*

$$\nu(\varnothing) = 500, \quad \nu(\{S\}) = 850, \quad \nu(\{L\}) = 850, \quad \nu(\{S, L\}) = 900.$$

*From these, the Möbius coefficients can be computed via Lemma 2:*

$$\mu(\{S\}) = 50, \qquad \mu(\{L\}) = 50, \qquad \mu(\{S, L\}) = 300.$$

*This leads to the following Shapley values (via Lemma 3):*

$$\psi_S = \mu(\{S\}) + \frac{1}{2}\mu(\{S, L\}) = 200, \qquad \psi_L = \mu(\{L\}) + \frac{1}{2}\mu(\{S, L\}) = 200.$$

*Now consider a modified model $f' : \mathbb{R}^3 \to \mathbb{R}$, where the size feature is duplicated, resulting in $X_{S_1}$ and $X_{S_2}$. The new model is defined as $f'(x_{S_1}, x_{S_2}, x_L) := f(x_S, x_L)$, where $x_{S_1} = x_{S_2} = x_S$, which leads to the following $\nu$ values (all the other are zero):*

$$\nu(\varnothing) = 500, \nu(\{S_1\}) = \nu(\{S_2\}) = \nu(\{L\}) = \nu(\{S_1, S_2\}) = 850, \nu(\{S_1, L\}) = \nu(\{S_2, L\}) = \nu(\{S_1, S_2, L\}) = 900.$$

*which leads to the Möbius coefficients via Lemma 2 (all the other are zero):*

$$\mu(\{S_1, S_2\}) = 50, \qquad \mu(\{L\}) = 50, \qquad \mu(\{S_1, S_2, L\}) = 300.$$

*As before, we compute the Shapley values for model $f'$. Lemma 3 gives:*

$$\psi_{S_1} = 125, \qquad \psi_{S_2} = 125, \qquad \psi_L = 150.$$

Adding a duplicate variable, which introduces no new information, causes the contribution of the location feature to decrease from 200 to 150. Moreover, further duplication of the size variable can reduce the location attribution even more, potentially down to 50. In such a case, the original entire joint contribution $\mu(\{S, L\})$ would be allocated exclusively to the duplicated size variables, marginalizing the location feature entirely. This paradox illustrates a fundamental flaw in standard Shapley-based explanations: duplicating a feature (without adding any new information) can unfairly inflate its contribution at the expense of others. Notably, this effect is not limited to perfect copies of features; it can also occur when adding statistical proxies or correlated duplicates. Moreover, while detecting perfect duplicates, statistical proxies, or correlated features is possible, the paradox can be present in more intricate cases involving redundancy of information across subsets of features, where detection is not easy, but still leads to problematic cases, as illustrated with the next example. Another example of this type, where a feature is redundant but a duplicate is presented, is presented in Appendix B.

## 5 Equal division vs Duplicate-feature paradox

Equal division is a fundamental property that is often desirable for feature attribution methods. While many approaches, including Shapley values, successfully satisfy this property, most fail to satisfy duplication-invariance. As highlighted in Definition 7, any method that resolves the duplicate-feature paradox in the presence of redundant variables, violates equal division by enforcing unchanged attributions for other features. An ideal attribution method would preserve equal division in instances that do not contain redundant features, and permit its violation only in the presence of redundant variables. In the following theorem, we show that no attribution method can satisfy equal division on all non-redundant instances while also satisfying duplication-invariance. In other words, these two properties are fundamentally incompatible.

**Theorem 9.** *There does not exist a duplication-invariant attribution method that satisfies the equal division property even for instances that do not contain redundant features.*

*Proof.* For contradiction, assume that there exists a duplicate invariance attribution method $\psi$ that satisfies equal division also on instances without redundant variables. Consider the two instances described in Observation 4. By construction, both instances contain no redundant features.

In the first instance (with features $A$, $B$, and $C$), the equal division property yields

$$\psi_C = \mu(\{C\}) + \frac{1}{3}\mu(\{A, B, C\}) = 2 + 4 = 6.$$

In the second instance (with features $C$, $D$), the same property gives

$$\psi_C = \mu(\{C\}) + \frac{1}{2}\mu(\{C, D\}) = 2 + 6 = 8.$$

Now imagine extending the first instance by adding the feature $D$ from the second instance, and extending the second instance by adding features $A$ and $B$ from the first. This results in both extended instances being identical. By duplication-invariance, the attribution for feature $C$ must remain unchanged in both instances. However, $\psi_C$ in the first instance and in the second are different, which is a contradiction. Thus, the only way to reconcile this contradiction is to allow that at least one of the instances violated the equal division property, even though both contained no redundant features. This contradicts our assumption, completing the proof. □

## 6 ReSHAP: A Redundancy-Weighted Generalization of Shapley Attribution

Several approaches have been proposed in the literature to address redundancy limitations of Shapley values, many of which involve modifying the weight vector, either directly in the form presented in Definition 1, or through its equivalent formulation using permutations; see, e.g., Frye et al. (2020); Kwon & Zou (2022). In this section, we build upon the reformulation of Shapley values presented in the *Reformulation of Shapley values* section.

The Shapley values introduced in Lemma 3 provide an explicit decomposition of contributions across all interaction terms between features. Compared to the standard permutation-based definition, this formulation allows us to precisely localize contributions to specific subsets of features. Leveraging this, we propose a new approach to compute redundancy-invariant Shapley values by introducing custom weight vectors at the level of the Möbius coefficients $\mu$, which operate directly on the atomic intersection structure of feature subsets. Since it introduces modifications directly at the level where the equal division property originates, this enables us to approach a solution to the duplicate-feature paradox; see Theorem 9.

Here, we propose a method that satisfies the redundancy-invariant property. From Theorem 9 we know that such a method must reconsider the equal division of atomic intersections of features even for the most basic instances like the one in Example 8. To better translate the theoretical results in Example 8 into a concrete attribution method, recall from Definition 5 that for any redundant feature $j$, there exists a set $S \subseteq [n] \setminus \{j\}$ such that $\nu(S \cup \{j\}) = \nu(S)$. According to duplication-invariance (Definition 7), we require that the attributions assigned to features outside of $S$ remain unchanged after adding the redundant feature $j$ and that the total attribution mass assigned to features in $S$ must remain the same before and after adding $j$. Intuitively, in the Venn diagram interpretation, adding feature $j$ does not increase the overall volume associated with the region defined by $S$; it only subdivides it into smaller regions that now include $j$. Our approach is to assign attribution to each of these subregions proportionally, based on their contribution to the original volume associated with $S$.

Compared to correlation-based approaches (see Appendix C), this method accounts for interactions between feature subsets of arbitrary cardinality. Redundancy is measured by comparing the volume of specific atomic regions in the Venn diagram to the overall volume of the region. More precisely to compute the fraction of attribution from $\mu(T)$ that is assigned to $i \in T$ we compute the independent contribution of $i$ for $T$ but also contributions of subsets of $T$ containing $i$ for $T$. Such subsets then further divide these contributions among their features until the final contribution for each individual feature is computed. For clarity of presentation, we describe our method recursively, highlighting how attribution within intersecting regions is progressively reallocated among features as redundancy is introduced.

We are ready to present the final definition of ReSHAP. Note that the computational complexity of ReSHAP is presented in Appendix F and the comparison with other methods is in Appendix G.

**Definition 10** (ReSHAP). *The ReSHAP method assigns to each feature $i \in [n]$ an attribution score*

$$\phi_i := \sum_{T \subseteq [n]} w_i(T) \cdot \mu(T),$$

*where for every $T \subseteq [n]$, $w_i(T) \in [0, 1]$ are feature-specific redistribution weights satisfying: $\sum_{i \in T} w_i(T) = 1$ where $w_i(T) = 0$ for $i \notin T$ and $w_i(T)$ for $i \in T$ are computed by the recursive redistribution procedure described in Algorithm 1.*

**Theorem 11.** *Let $f : \mathbb{R}^n \to \mathbb{R}$ be a model function, and let $f' : \mathbb{R}^{n+1} \to \mathbb{R}$ be a model extension in which the feature $x_{n+1}$ is redundant. A ReSHAP feature attribution function $\phi_i \in \mathbb{R}$ for $i \in [n+1]$, is duplication-invariant.*

---

**Algorithm 1** Recursive Redistribution of Möbius Mass for ReSHAP

---

1: For a fixed $T$, initialize $w_i(T) := 0$ for all $i \in [n]$
2: call Distribute($S = T, \mu(T)$)
3: **procedure** DISTRIBUTE($S$, mass)
4:    **if** $|S| = 1$ **then**
5:       let $i$ be the unique element of $S$
6:       $w_i(T) \leftarrow w_i(T)+$ mass
7:       **return** $w_i(T)$
8:    **else**
9:       **if** $\sum_{\varnothing \neq V \subset S} |\mu(V)| = 0$ **then**
10:          **for all** $i \in S$ **do**
11:             $\xi(\{i\}) \leftarrow \frac{1}{|S|}$
12:             DISTRIBUTE($\{i\}, \xi(\{i\})\cdot$ mass)
13:          **end for**
14:       **else**
15:          **for all** non-empty $U \subset S$ **do**
16:             $\xi(U) \leftarrow \dfrac{|\mu(U)|}{\sum_{\varnothing \neq V \subset S} |\mu(V)|}$
17:             DISTRIBUTE($U, \xi(U)\cdot$ mass)
18:          **end for**
19:       **end if**
20:    **end if**
21: **end procedure**

---

*Proof.* We want to prove that

$$\phi_i^{f'} = \phi_i^f \qquad \text{for all } i \in [n] \setminus S$$
$$\sum_{k \in S \cup \{n+1\}} \phi_k^{f'} = \sum_{k \in S} \phi_k^f$$

where $S \subseteq [n]$ is a minimal set satisfying the condition in Definition 5.

We start with proving the second one. Indeed, note that, since for the redundant feature $n + 1$ we have $\nu(S \cup \{n + 1\}) = \nu(S)$ it implies that $\mu(\{n + 1\}) = 0$ (it provides no new information to the system). Thus in the recursive redistribution, if $\sum_{V \subset T} \mu(V) \neq 0$ feature $n + 1$ gets no mass assigned, that is $\xi(\{n + 1\})$ is always zero since the numerator $\mu(\{n + 1\})$ is zero. On the other hand if $\sum_{\varnothing \neq V \subset S} |\mu(V)| = 0$ the mass is equally distributed among the features in $S$ so the redundant feature gets mass only at the cost of other features from $S$. This implies that all the mass from features in $S$ after adding redundant feature $n + 1$ goes again to features in $S$ so it is preserved. To prove the first condition, given that the second one holds, it suffices to argue that for all $T \subseteq [n]$ every recursive step in Algorithm 1 redistributes mass $\mu(T)$ only among features $i \in T$, and preserves it. So the remaining mass for features outside $S$ goes with the same amount before and after adding feature $n + 1$ and it is redistributed proportionally to their individual mass $\mu$ thus stays unchanged after adding feature $n + 1$. $\qquad\square$

## 7 ANALYSIS AND EMPIRICAL EVALUATION

### 7.1 CASE STUDIES ON SYNTHETIC EXAMPLES

We revisit the two running examples introduced earlier in Example 8 and B (In Appendix D) and compute ReSHAP attributions alongside standard Shapley values. This illustrates how the recursive redistribution alters feature importance in the presence of duplicates and redundant features.

We start with Example 8. Recall that after adding a duplicate feature of $S$ we had the following values of $\mu$ (all others $= 0$):

$$\mu(\{L\}) = 50, \ \mu(\{S_1, S_2\}) = 50, \ \mu(\{S_1, S_2, L\}) = 300.$$

Now for every nonzero value of $\mu(T)$ we call Algorithm 1 to compute weights $w_i(T)$ for all features.

- We start with $T = \{L\}$: since $T$ is a singleton, Algorithm 1 assigns $w_L(\{L\}) = 50$.

- For $T = \{S_1, S_2\}$, since $\mu(S_1) = \mu(S_2) = 0$, the algorithm splits equally $\mu(\{S_1, S_2\})$, yielding $w_{S_1}(\{S_1, S_2\}) = w_{S_2}(\{S_1, S_2\}) = 25$.

- Finally, for $T = \{S_1, S_2, L\}$ the algorithm proportionally splits the mass of 300 with $\xi(\{S_1, S_2\}) = \xi(\{L\}) = 0.5$, which in the recursive call gives $w_{S_1}(\{S_1, S_2, L\}) = w_{S_2}(\{S_1, S_2, L\}) = 75$ and $w_L(\{S_1, S_2, L\}) = 150$.

This, by Definition 10, leads to ReSHAP values:

$$\phi_{S_1} = \phi_{S_2} = 100, \quad \phi_L = 200,$$

which coincides with the distribution of $\psi$ values before adding a duplicate, where contribution of $S$ was split between $S_1$ and $S_2$.

ReSHAP performance for the second example from Appendix B is presented in Appendix D.

## 7.2 SMALL-SCALE REAL DATA EXPERIMENT

To provide a proof of concept, we conducted a small experiment (the full experiment can be found in Appendix E) on the well-known *Ames Housing* dataset, which contains detailed information on 79 explanatory variables related to residential properties. The dataset includes both numerical and categorical variables, covering a broad range of structural and qualitative characteristics of the houses. The target variable is the `SalePrice`, representing the sale price of the properties. For our experiment, we focus on three key features that are both meaningful and highly predictive: the above-ground living area (`Gr Liv Area`), the overall quality of the house (`Overall Qual`), and the total number of rooms above grade (`TotRms AbvGrd`).

A Multi-Layer Perceptron (MLP) model was trained on this dataset. Feature attribution was performed using both standard SHAP values and our proposed ReSHAP values, which distribute prediction contributions across input features. To evaluate explanation robustness, we conducted controlled experiments in which we introduced redundancy either by duplicating an existing feature or by including a correlated feature. The goal was to examine how stable SHAP and ReSHAP attributions remain when redundancy is present. Stability was quantified using diagnostic measures $P$ and $R$, which capture the change in relative importance of a non-redundant feature before and after introducing redundancy, measured respectively with SHAP and ReSHAP values. Their ratio $P/R$ serves as a stability indicator: values close to one indicate similar behavior between SHAP and ReSHAP, whereas larger values reveal instability in SHAP that ReSHAP successfully mitigates.

The results, summarized in Table 3, show clear differences between the two modes. In the duplicate-feature setting (`dup_qual`), SHAP exhibited more instability compared to ReSHAP, with a mean $|P/R|$ of 5.49 across 100 test points, reflecting large shifts in feature importance. In the correlated-feature setting (`totrms`), the instability was smaller but still present, with a mean $|P/R|$ of 2.53. Once again, ReSHAP produced more consistent explanations. These findings confirm that ReSHAP provides a more robust and reliable attribution framework, even in real-world data.

## 8 CONCLUSIONS AND FUTURE WORK

In this paper, we introduced ReSHAP, a redundancy-aware generalization of Shapley attribution that resolves the duplicate-feature paradox while retaining the canonical value function and compatibility with existing SHAP frameworks. There are many promising avenues for future work. A first direction is to optimize the computation of exact and approximate ReSHAP values, for example by leveraging TreeSHAP and KernelSHAP techniques to accelerate the underlying Shapley evaluations. Further experimental validation on larger benchmarks, as well as extensions to other domains such as time series data, also represent important directions for future research.

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

USAGE OF LLMs IN PAPER

During the preparation of this work, we made use of the large language model ChatGPT (OpenAI, GPT-5) to support two aspects of the research. First, it was employed to improve the readability and style of the manuscript by rephrasing draft passages into clearer, more concise academic text. Second, it was used to assist in the implementation of the computational experiments, for example by suggesting code fragments, debugging strategies, and formatting options for tables and figures. All conceptual contributions, experimental design decisions, and interpretation of the results remain the responsibility of the authors. The use of ChatGPT was limited to supporting tasks, and the scientific content, analysis, and conclusions of this paper were produced entirely by the authors.

## A  PROOF OF LEMMA 3

We start with plugging in formulation from Lemma 2 into the definition of Shapley values to get

$$\psi_i = \sum_{S \subseteq [n] \setminus \{i\}} \frac{|S|!(n-|S|-1)!}{n!} \sum_{\substack{T \supseteq \{i\} \\ T \cap S = \varnothing}} \mu(T) = \sum_{S \subseteq [n] \setminus \{i\}} \sum_{\substack{T \supseteq \{i\} \\ T \cap S = \varnothing}} \frac{|S|!(n-|S|-1)!}{n!} \mu(T)$$

$$= \sum_{T \supseteq \{i\}} \sum_{\substack{S \subseteq [n] \setminus \{i\} \\ S \cap T = \varnothing}} \frac{|S|!(n-|S|-1)!}{n!} \mu(T) = \sum_{T \supseteq \{i\}} \mu(T) \sum_{k=0}^{n-|T|} \binom{n-|T|}{k} \frac{k!(n-k-1)!}{n!}$$

where, after the change of variables $r = n - |T| - k$ we get

$$\sum_{k=0}^{n-|T|} \binom{n-|T|}{k} \frac{k!(n-k-1)!}{n!} = \frac{(n-|T|)!}{n!} \sum_{k=0}^{n-|T|} \frac{(n-k-1)!}{(n-|T|-k)!} = \frac{(n-|T|)!}{n!} \sum_{r=0}^{n-|T|} \frac{(r+|T|-1)!}{r!},$$

which is equal to $\frac{1}{|T|}$, see Lemma 12.

**Lemma 12.** *For every $n \in \mathbb{N}$ and $m \leq n$ it holds:*

$$\sum_{r=0}^{n-m} \frac{(r+m-1)!}{r!} = \frac{n!}{(n-m)!} \frac{1}{m}.$$

*Proof.* Indeed, using the Pascal identity for binomial coefficients we know that

$$\binom{r+m}{r} = \binom{r+m-1}{r-1} + \binom{r+m-1}{r},$$

which implies the following identity

$$\frac{(r+m)!}{r!} - \frac{(r+m-1)!}{(r-1)!} = m \frac{(r+m-1)!}{r!}.$$

Plugging into the original summation yields

$$\sum_{r=0}^{n-m} \frac{(r+m-1)!}{r!} = \frac{1}{m} \sum_{r=0}^{n-m} \left( \frac{(r+m)!}{r!} - \frac{(r+m-1)!}{(r-1)!} \right) = \frac{1}{m} \frac{n!}{(n-m)!}$$

where the last equality holds since the expression in the summation is a telescopic sum where all the elements cancel out except of the term $\frac{(r+m)!}{r!}$ for $r = n - m$.

$\square$

## B EXAMPLE FOR SHAPLEY VALUES FOR REDUNDANT NONDUPLICATED FEATURE

Let $f : \mathbb{R}^3 \to \mathbb{R}$ be a model function that estimates the risk of credit default based on a random vector with three components: annual income $X_A$, debt-to-income ratio $X_B$, and credit score $X_C$ (all understood as real numbers). Let $x_A$, $x_B$, and $x_C$ be specific values under which the model predicts the risk.

Assume the following values of the $\nu$ function, for the sake of simplicity we assume $\nu(\varnothing) = 0$:

$$\nu(\{A\}) = 14 \qquad \nu(\{C\}) = 14 \qquad \nu(\{A, B\}) = 16 \qquad \nu(\{A, B, C\}) = 18$$
$$\nu(\{B\}) = 14 \qquad \nu(\{A, C\}) = 16 \qquad \nu(\{B, C\}) = 16$$

The Möbius coefficients can be computed via Lemma 2 (all others= 0), see also Figure 2 on the left:

$$\mu(\{A\}) = 2, \qquad \mu(\{B\}) = 2, \qquad \mu(\{C\}) = 2, \qquad \mu(\{A, B, C\}) = 12.$$

This leads to the Shapley values (via Lemma 3):

$$\psi_A = \mu(\{A\}) + \frac{1}{3}\mu(\{A, B, C\}) = 2 + \frac{1}{3} \cdot 12 = 6,$$

$$\psi_B = \mu(\{B\}) + \frac{1}{3}\mu(\{A, B, C\}) = 2 + \frac{1}{3} \cdot 12 = 6,$$

$$\psi_C = \mu(\{C\}) + \frac{1}{3}\mu(\{A, B, C\}) = 2 + \frac{1}{3} \cdot 12 = 6.$$

Now consider a modified model $f' : \mathbb{R}^4 \to \mathbb{R}$, where a new feature, total monthly loan payment $X_D$, is added. This feature is partially determined by $X_A$ and $X_B$, lying in their algebraic span, but it is neither a copy of $X_A$ nor of $X_B$. It adds information to $X_A$ and $X_B$ individually but adds no new information beyond their joint contribution.

In this case, suppose the value function $\nu$ takes the following values:

$$\nu(\{A\}) = 14 \quad \nu(\{D\}) = 14 \quad \nu(\{A, D\}) = 15 \quad \nu(\{C, D\}) = 16 \quad \nu(\{A, C, D\}) = 17$$
$$\nu(\{B\}) = 14 \quad \nu(\{A, B\}) = 16 \quad \nu(\{B, C\}) = 16 \quad \nu(\{A, B, C\}) = 18 \quad \nu(\{B, C, D\}) = 17$$
$$\nu(\{C\}) = 14 \quad \nu(\{A, C\}) = 16 \quad \nu(\{B, D\}) = 15 \quad \nu(\{A, B, D\}) = 16 \quad \nu(\{A, B, C, D\}) = 18$$

The corresponding Möbius coefficients are (all others= 0), see also Figure 2 on the right:

$$\mu(\{A\}) = 1 \qquad \mu(\{C\}) = 2 \qquad \mu(\{B, D\}) = 1$$
$$\mu(\{B\}) = 1 \qquad \mu(\{A, D\}) = 1 \qquad \mu(\{A, B, C, D\}) = 12$$

As before, using Lemma 3, the Shapley values for $f'$ are:

$$\psi_A = \mu(\{A\}) + \frac{1}{2}\mu(\{A, D\}) + \frac{1}{4}\mu(\{A, B, C, D\}) = 4.5,$$

$$\psi_B = \mu(\{B\}) + \frac{1}{2}\mu(\{B, D\}) + \frac{1}{4}\mu(\{A, B, C, D\}) = 4.5,$$

$$\psi_C = \mu(\{C\}) + \frac{1}{4}\mu(\{A, B, C, D\}) = 5,$$

$$\psi_D = \frac{1}{2}\mu(\{A, D\}) + \frac{1}{2}\mu(\{B, D\}) + \frac{1}{4}\mu(\{A, B, C, D\}) = 4.$$

Again, we observe that adding a new variable, although not a copy of any existing variable or a subset thereof, and contributing no new information, leads to a decline in the value of feature $C$ from 6 to 5. Further addition of similar variables could reduce the attribution of $C$ even further, potentially down to its standalone contribution of 2.

## C POTENTIAL OF USING CORRELATIONS TO ADDRESS REDUNDANCY

In this subsection, we consider simpler measures for feature dependence such as correlation. For instance, if two features have a high Pearson correlation, we might consider them partly redundant

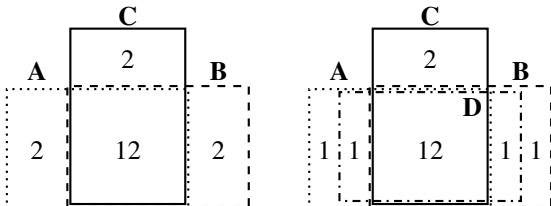

Figure 2: Venn diagrams showing $\mu$ values for Example B for models with 3 features (left) and 4 features (right).

and reduce the individual attributions accordingly. Indeed, some earlier heuristics for feature importance recommend grouping or discounting highly correlated features Zhao et al. (2019). However, using correlations alone to *solve* for the duplicate-feature problem has its limitations.

First, correlation typically measures pairwise dependency, capturing relationships between two variables at a time. However, feature redundancy can involve higher-order interactions that pairwise correlations miss. For instance, two features may be uncorrelated with the target individually but informative together (e.g., XOR). Similarly, features may appear uncorrelated yet exhibit redundancy when considered with a third variable (see Example B). Thus, pairwise correlation is insufficient to detect such dependencies.

Secondly, even with pairwise relationships, correlation is symmetric and offers no guidance on attribution between correlated features. For example, if features $A$ and $B$ correlate at $0.9$, how should their Shapley values be split? Arbitrary adjustments risk violating Shapley axioms or producing inconsistent explanations. That said, some recent works have implicitly used correlation information to refine explanations. For example, Merrick & Taly (2020) use cohort clustering to reduce feature dependence by grouping instances with lower within-group correlations. Another exception is Aas et al. (2021), who generalize Kernel SHAP by replacing independence assumptions with data-driven conditional sampling, implicitly leveraging feature correlations for more accurate attributions. Finally, Basu & Maji (2021) apply a linear adjustment using the covariance matrix to account for feature dependencies via linear correlations. These approaches indicate that incorporating correlation can improve attribution fairness, but they also reveal the limitations we discussed: the linear adjustment proposed by Basu & Maji (2021), for example, guarantees that attributions no longer depend on feature correlations in a linear sense, but it assumes the relationships are well-modeled by covariance (a Gaussian assumption) and doesn't directly extend to non-linear dependencies.

## D RESHAP PERFORMANCE FOR EXAMPLE B

For Example B (four features after adding $D$), the nonzero Möbius values are (all others $= 0$):

$$\mu(\{A\}) = \mu(\{B\}) = 1, \ \mu(\{C\}) = 2, \ \mu(\{A, D\}) = \mu(\{B, D\}) = 1, \ \mu(\{A, B, C, D\}) = 12.$$

Apply Algorithm 1 to each nonempty $T$.

- For all the singletons we get directly $w_A(\{A\}) = 1$, $w_B(\{B\}) = 1$, $w_C(\{C\}) = 2$.

- For $T = \{A, D\}$, since $|\mu(\{A\})| = 1$ and $|\mu(\{D\})| = 0$, the mass $\mu(\{A, D\}) = 1$ goes to $A$, so $w_A(\{A, D\}) = 1$, $w_D(\{A, D\}) = 0$.

- Analogously for $T = \{B, D\}$ we get $w_B(\{B, D\}) = 1$, $w_D(\{B, D\}) = 0$.

- For $T = \{A, B, C, D\}$, the proper nonempty subsets with nonzero $\mu$ are $\{A\}, \{B\}, \{C\}, \{A, D\}, \{B, D\}$ with magnitudes $1, 1, 2, 1, 1$ (sum $= 6$), hence $\xi(\{A\}) = \xi(\{B\}) = \xi(\{A, D\}) = \xi(\{B, D\}) = \frac{1}{6}$ and $\xi(\{C\}) = \frac{2}{6}$.

  Recursing: mass $12 \cdot \frac{1}{6} = 2$ to adds 2 to $A$ and similarly adds 2 to $B$. Mass $12 \cdot \frac{2}{6} = 4$ adds 4 to $C$; mass $12 \cdot \frac{1}{6} = 2$ to $\{A, D\}$ all goes to $A$; and $12 \cdot \frac{1}{6} = 2$ to $\{B, D\}$ all goes to $B$.

  Thus $w_A(\{A, B, C, D\}) = 4$, $w_B(\{A, B, C, D\}) = 4$, $w_C(\{A, B, C, D\}) = 4$, $w_D(\{A, B, C, D\}) = 0$.

By Definition 10, the ReSHAP attributions are: $\phi_A = 1 + 1 + 4 = 6$, $\phi_B = 1 + 1 + 4 = 6$, $\phi_C = 2 + 4 = 6$, $\phi_D = 0$, which is the same the original attribution before adding feature $D$. Note that feature $D$ gets zero attribution as it not only does not bring any new information, but is a strict subset of features $A \cup B$.

# E  SMALL-SCALE REAL DATA EXPERIMENT

To provide a proof of concept, we include a small experiment on a real-world dataset[1] In this experiment, we investigate whether SHAP explanations align with established economic intuition and how correlated features influence interpretability. Our contribution in this experiment is the option to benchmark SHAP against ReSHAP. While not exhaustive, this demonstrates the practical feasibility of ReSHAP and its behavior compared to baseline methods. Key features in this experiment include:

- `Gr Liv Area`: Above-ground living area in square feet.
- `Overall Qual`: Overall material and finish quality of the house.
- `TotRms AbvGrd`: Total rooms above grade.
- `SalesPrice`: Target variable.

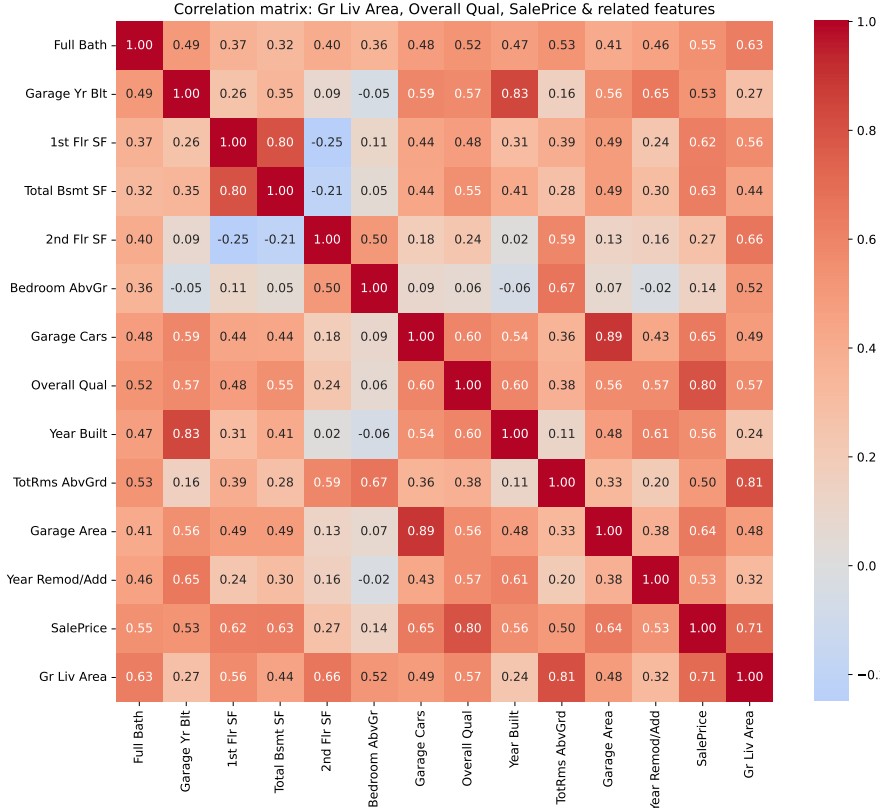

Figure 3: Correlation matrix in the Ames Housing dataset including `Gr Liv Area`, `Overall Qual`, `SalePrice`, and related features.

We train a Multi-Layer Perceptron (MLP) model on the housing dataset with the hyperparameters given in Table 1.

---

[1]The Ames Housing dataset provides a dataset with 79 explanatory variables related to properties (see Figure 3 for an overview of some of the features in the dataset). The target variable in the dataset is the `SalePrice`, representing the sale price of the houses. The features include both numerical and categorical variables, covering a wide range of aspects such as lot size, number of rooms, location, construction, and more.

Table 1: MLP model and its hyperparameters.

| Model | Hyperparameters |
| --- | --- |
| MLP Regressor | Pipeline: `("scaler", StandardScaler())` $\rightarrow$ `("model",` `MLPRegressor(...))`. `hidden_layer_sizes = (64, 32)`, `activation = "relu"`, `solver = "adam"`, `max_iter = 5000`, `early_stopping = True`, `n_iter_no_change = 20`, `tol = 1e-4`, `learning_rate_init = 1e-3`, `alpha = 1e-4`, `random_state = 42`. |

Feature attribution is performed using SHAP and ReSHAP values, which distribute prediction contributions across input features. To evaluate the differences between both algorithms, we conduct controlled experiments by introducing duplicate or correlated features. Our evaluation follows four steps:

1. Correlation analysis to identify strongly related features.
2. Computation of predictive performance of the MLP model used (e.g., RMSE and $R^2$)
3. Computation of both SHAP and ReSHAP attributions.
4. Analysis of SHAP and ReSHAP attributions, with special focus on the duplicate or correlated feature.

To study explanation robustness, we randomly select 10 test points (`N_RANDOM_POINTS=10`) and compute SHAP and ReSHAP values for each point. The baseline for $f(\emptyset)$ is chosen as the training-set mean prediction, ensuring that all contributions are measured relative to an intuitive baseline. For each model and test point, we compute:

- *Subset predictions* for all feature subsets,
- *SHAP values* ($\phi$) per feature,
- *Lattice functions* $f(S)$ for subsets $S$,
- *Interaction indices* ($\mu$), and
- *ReSHAP values*, obtained via recursive attribution.

This evaluation allows us to assess not only global feature importance but also how explanations behave under redundancy and multicollinearity. In particular, duplicate-feature scenarios (`dup_grliv`, `dup_qual`) reveal whether SHAP splits contributions equally or arbitrarily, while the correlated-feature scenario (`totrms`) tests the method's ability to extract overlapping effects. The experimental settings are shown in Table 2. The mode gives us the option to vary within modes. For this experiment, `dup_qual` and `totrms` are chosen.

Table 2: Key experiment settings used in the pipeline.

| Setting | Value |
| --- | --- |
| Feature mode (`MODE`) | `"two"`, `"dup_grliv"`, `"dup_qual"`, `"totrms"`: uses either a duplicate of *Gr Liv Area* or *Overall Qual*, or a correlated feature *TotRms AbvGrd*. |
| Baseline for $f(\emptyset)$ | `"mean"` |
| Test-point selection | `PICK_RANDOM_POINT=True` |
| # test points assessed | `N_RANDOM_POINTS=100` |
| Model(s) enabled | MLP |

To evaluate the stability of feature attributions under redundancy, we introduce diagnostic measures $P$, $R$, and their ratio $P/R$. The measure $P$ is defined as the difference (in percentage points) between the absolute relative importance of a non-redundant feature before adding a redundant feature and

the absolute relative importance of the same non-redundant feature after the redundant feature has been added, both computed using standard Shapley values. In other words, $P$ captures how much the attribution of a non-redundant feature changes under SHAP when redundancy is introduced. The measure $R$ is defined in the same way, but using ReSHAP values: it is the difference (in percentage points) between the absolute relative importance of a non-redundant feature before and after adding a redundant feature, computed with ReSHAP. Thus $R$ quantifies the stability of ReSHAP under redundancy.

The ratio $P/R$ serves as a comparative stability indicator: values close to one suggest that SHAP and ReSHAP behave similarly, whereas values significantly larger than one highlight cases where SHAP attributions fluctuate stronglyafter adding a redundant feature while ReSHAP remains stable.

SUMMARY OF RESULTS

Table 3: Overview of mean absolute $|P/R|$ values for the MLP model across two selected experimental modes. Higher values indicate greater instability of SHAP relative to ReSHAP. Results are averaged over multiple random test points.

| Experimental mode | mean($|P/R|$) | Number of test points |
|---|---|---|
| Duplicate-feature mode (`dup_qual`) | 5.49 | 100 |
| Correlated-feature mode (`totrms`) | 2.53 | 100 |

**Duplicate-feature mode.** When a duplicate of the *Overall Quality* feature is introduced in the `dup_qual` setting, SHAP becomes highly unstable in how it allocates importance between the original and the duplicate. This is reflected in a large mean $|P/R|$ value of $5.49$ across 100 random test points, indicating that the change in relative importance under duplication is far greater for SHAP than for ReSHAP. In other words, ReSHAP maintains more stable attributions for the non-redundant feature, while SHAP exhibits larger deviations. This finding highlights the corrective role of ReSHAP in the presence of duplicated variable.

**Correlated-feature mode.** When including *TotRms AbvGrd*, which is correlated with *Gr Liv Area* (as seen in Figure 3), the instability of SHAP is reduced compared to the duplication case, but is still notable. The mean $|P/R|$ value of $2.53$ across 100 test points indicates that SHAP explanations shift substantially in response to correlation, while ReSHAP again provides more stable attributions. Although the divergence is smaller than in the duplicate-feature setting, the results confirm that correlation alone is sufficient to destabilize SHAP attributions, whereas ReSHAP mitigates this effect.

## F  COMPUTATIONAL COMPLEXITY AND PRACTICAL CONSIDERATIONS

Although the results in the paper contribute on the fundamental side of cooperative game theory and its connection to Shapley values, below we analyze the computational cost of ReSHAP relative to standard Shapley estimation. We also discuss implementation aspects, including compatibility with approximation schemes such as KernelSHAP and TreeSHAP, and potential speed-ups. The computational complexity of computing Shapley values exactly in the worst case is known to be exponential, i.e., $O(n2^n)$. Since our work makes a contribution on the fundamental principle of how the Shapley values are constructed, solving the duplicate feature paradox, its exact computation should not be expected to be smaller than Shapley values itself. In fact, it is very comparable from a computational complexity perspective.

Indeed, the algorithm to compute ReSHAP requires first computation of Shapley values $\nu$ in time $O(2^n)$. Then given $\nu$ computing value $\mu(T)$ for subset $T$ of cardinality $k$, naively, requires summing up $2^k$ elements, thus total number of operations required to compute $\mu$ for all $T \subseteq [n]$ is $\sum_{i=0}^{k} \binom{n}{k} 2^k = O(3^n)$. Using zeta/Möbius transform drops it to $O(n2^n)$ Finally, the computational complexity of computing ReSHAP requires computing weights $w_i(T)$. Note that for a fixed subset $T \subseteq [n]$, Algorithm 1 computes all values $w_i(T)$ for all $i \in [n]$ by recursively calling procedure

DISTRIBUTE. For fixed $T$ of cardinality $k$ the procedure recurses on its subsets in a way that each of subsets $S \subseteq T$ of cardinality $\ell$ is called $2^{k-\ell}$ times. In each recursion the only computationally heavy component is computing $\sum_{\varnothing \neq U \subset S} |\mu(U)|$ in time $O(2^\ell)$. Thus computational complexity of recurring the procedure DISTRIBUTE for a fixed set $T$ of cardinality $k$ is $\sum_{\ell=0}^{k} \binom{k}{\ell} 2^{k-\ell} 2^\ell = O(4^k)$. Since we call it for each subset the overall complexity is $\sum_{k=0}^{n} \binom{n}{k} O(4^k) = O(5^n)$. The combined complexity of computing ReSHAP is $O(2^n + n2^n + 5^n) = O(5^n)$, which can be written as $O(2^{n \log 4}) = O(2^{2.33n})$. Although increased compared to the exact computation of Shapley values, still both are exponential time and differ by a small multiplicative constant in the exponent.

It is worth pointing out that several speed-ups are possible, both for computing exact ReSHAP values and its approximations. First, caching the values of $\sum_{\varnothing \neq U \subset S} |\mu(U)|$ instead of recomputing them every time the subset is called could already improve the computational complexity of recurring the procedure DISTRIBUTE. Moreover, Algorithm 1 can be optimized by computing weights for all subsets simultaneously, by first distributing the mass of $[n]$ into its subsets, but then recursively calling only subsets of cardinality one less, which also distribute their mass to their subsets of cardinality one less, etc. This could further reduce the overall complexity to $O(2^n)$, leading to an overall complexity of ReSHAP to $O(3^n)$. Finally, it is important to point out that this research does not aim to provide an optimized method for practical use, rather provides contribution on a fundamental level where the exponential time procedure of computing Shapley values is replaced with another exponential time procedure, ReSHAP, that has provably better behaviour in the presence of duplicate or redundant features. However, we would like to point out that several methods readily available to speed up exact and approximate computation of Shapley values could be applied to speed up computation of ReSHAP, mostly because its core is based solely on values $\nu$ and does not need extra knowledge of probability distributions or other statistics of the data. This includes techniques like TreeSHAP Lundberg et al. (2020) for tree based models, or Kernel SHAP Lundberg & Lee (2017).

## G COMPARISON WITH EXISTING APPROACHES

In this section, we give a short comparison of the ReSHAP method for solving the Duplicate-feature paradox with other existing methods, such as Frye et al. (2020); Kwon & Zou (2022); Watson et al. (2023); Ay et al. (2020).

Although results such as those in Frye et al. (2020); Kwon & Zou (2022) can resolve the duplicate feature paradox for certain choices of weight vectors, they lack a universal principle for selecting weights. In contrast, our method provides a provable procedure to compute a vector of weights that resolves the duplicate feature paradox. On the other hand, unlike approaches in Watson et al. (2023) and Ay et al. (2020), which use KL divergence and mutual information to define new value functions, we retain the standard value function in the first step and subsequently apply redundancy measure to account for feature dependencies. Moreover, although modifying the value function in Watson et al. (2023); Ay et al. (2020) accounts for feature correlations, they do not modify the permutation weight vector, thus not guaranteeing a solution to the duplicate-feature paradox. Finally, their methods assume extra information about probability distributions to compute KL measures, which limits the practical applicability of the methods. Instead, our method builds upon the canonical value function, a cornerstone of Shapley value formulations, thus our approach is directly compatible with existing approximation techniques such as KernelSHAP: one can incorporate our redundancy-aware weighting into those algorithms, benefiting from their efficiency while fixing the credit allocation issue. To our knowledge, this is the first technique that fully addresses feature redundancy in Shapley explanations without altering the model or requiring heavy computations beyond the standard Shapley values estimation.

