# OpenReview forum: "Resolving the Duplicate-Feature Paradox with ReSHAP: A Redundancy-Weighted Generalization of Shapley Attribution"
_ICLR.cc/2026/Conference — ICLR 2026 Conference Withdrawn Submission_

### Official Review · Reviewer_CnPx · 2025-10-27

**Soundness:** 2
**Presentation:** 1
**Contribution:** 2
**Rating:** 2
**Confidence:** 4

**Summary:**

The paper presents ReSHAP as a new method to compute Shapley Values to compute Shapley Values while addressing what the authors coin the duplication paradox. The duplication paradox is presented as a model that receives the same input feature twice and then assigns different attribution scores to the remainder of the features compared to a model that receives the feature without duplication. To solve this, the paper presents then ReSHAP which redistributes the möbius coefficients onto coalitions that contain duplicated features.

**Strengths:**

- Shapley Values and attributions are very important!
- The topic of attribution methods' dependence on the underlying feature distributions and potentially un-intuitive explanations is an interesting research direction.

**Weaknesses:**

- **Contrived Problem:** The paper's contribution addresses a rather contrived problem in my opinion. I have never heard of a "duplication paradox" as the authors present it (lines 64-68) and in my opinion this is a non-problem for Shapely. If you have one model $f_1(x_1, ...,x_n)$ with $n$ features and another model  $f_2(x_1,x_1,...,x_n)$ operating on $n+1$ features, then this is seems to be a modeling issue if the model gets access to the same information in two of its input features and b) totally justified for the Shapley Values to be different for both models $f_1$ and $f_2$. While I would not directly know how the Shapley Values behave as this is extremely dependent on the model architecture (chosen hypothesis space), I do not see a Shapley issue here. Yet, this instantiation is related to correlated features and implications of the removal mechanism selected in the value function for computing the Shapley Values. However, this is a quite broad topic explored in many related works.
- **Very shallow Empirical Evaluation:** The empirical evaluation of this work is basically not existent. The only thing done is a Jupyter notebook containing case studies on synthetic data and really small datasets. Further no results are placed in the main body of the paper but referred to in the supplement. Thereby the empirical evaluation is not convincing at all. The *"problem"* is not properly motivated and the proposed ReSHAP does not show clear improvements compared to established baselines.
- **Writing and Presentation:** The paper's writing and presentation is not good. The work contains multiple technical examples that are not well placed in the text. The manuscript reads quite jarring and lacks a good structure. Proofs, proof sketches, and examples make up the majority of the methodological section which all take away the space for actually fleshing out the proposed problem more and addressing the core methodological contribution.
- **Unclear:**  It is unclear to me if the method needs to evaluate the power set of all coalitions or if it can be used in a estimation procedure akin to traditional Shapley value estimation methods (such as KernelSHAP, SVARM, or Permutation Sampling). The former is quite a big problem limiting the applicability of ReSHAP drastically and would require proposing an estimation method (and properly testing it) the and the latter still needs to do a proper evaluation of the estimation quality.
- **Missing Ablations.** The paper does not discuss or analyze ReSHAP's behavior on common parameters such as feature-size or hyper parameters (the above mentioned estimation budget).

**Questions:**

- What is the computational complexity of ReSHAP? Can I call it with a fixed budget for estimation purposes or do I need to evaluate the whole powers of coalitions?

---

> ### Author Response · Authors · 2025-11-26
>
> We would like to thank the Reviewer for their careful evaluation of the paper and for the remarks that may help improve it in the future. Given the preliminary rating, we must realistically assess the likelihood of acceptance. Nevertheless, we would like to respond to the reviewers’ questions, out of full respect for their work and to provide the requested clarifications. We would appreciate the opportunity for discussion and feedback in order to better understand how to strengthen the work and address the current gaps.
>
> In the following, we address the Reviewer’s comments in detail:
>
> **Contrived Problem:**
>
> We thank the Reviewer for this perspective. However, we do not agree with the argument. The paradox does not compare two different models; the original and duplicated models compute exactly the same function, so Shapley values should be identical for non-duplicate features.
> The issue is not that the duplicate receives some credit, but that non-duplicate features are penalized, violating basic invariance principles (Theorem 11 in our paper).
>  It is worth mentioning that the problem of adding a duplicated feature (or highly correlated or redundant feature from the perspective of the new information it brings to the model) was recognized as a main limitation of Shapley values in Kumar et. al. (ICML 2020). The problem is not “contrived.”
> Our result is a rigorous impossibility theorem showing that equal division forces this pathology, independent of model architecture, contrary to the reviewer’s belief.
>
>
> **Very shallow Empirical Evaluation**
>
> Although it is hard to disagree with this observation, we would like to emphasize the strong fundamental contribution of our result, which solves a well-known problem of Shapley values. Moreover, the current computational complexity makes it difficult to conduct extensive experimental validation. However, this limitation can be addressed in future work by following the line of research established for SHAP and translating existing approximation techniques to the ReSHAP setting (see Appendix F).
> We agree with the Reviewer that greater visibility of the experimental results would be beneficial for the paper, and we will consider moving (and possibly expanding, within the available possibilities) the experimental section into the main body of the paper.
>
> **Writing and Presentation**
> We thank the Reviewer for the honest feedback regarding the writing quality. We would also like to note that another reviewer (PiXE) appreciated the quality of the writing. Nevertheless, we will make an extra effort in the future to further improve the quality of the writing.
>
> **Unclear [...] What is the computational complexity of ReSHAP?**
> We thank the reviewer for asking this very important question. As presented in Appendix F, it is exponential in n. However, it is important to point out that this work aims to investigate the fundamental limitations of Shapley values, which, in their pure form, also require exponential computations in n. Thus, a foundational improvement to the Shapley value framework is a necessary step toward resolving its inherent limitations and an exponential running time is expected therein.
>
> That said, it is worth emphasizing that the machinery presented in this paper is not purely theoretical. A key component of our approach is that it relies solely on evaluations of the set function \nu. As discussed in Appendix F, such as PermutationSHAP, KernelSHAP, and related methods, may potentially be potentially adapted to yield efficient procedures capable of resolving the duplicate-feature paradox. Due to the depth of this research direction, we leave these developments for future work.
>
> **Missing Ablations.**
>
> We thank the reviewer for raising this point and the opportunity to clarify. ReSHAP does not introduce any new hyperparameters beyond those already present in the underlying SHAP estimators. All parameters that affect the numerical estimates of attributions are inherited entirely from these existing SHAP implementations; the ReSHAP transformation itself has no tunable knobs.
>
> More precisely, the ReSHAP mapping is defined by a fixed linear transform to the $\mu$-basis (Lemma 2) followed by Algorithm 1, which redistributes $\mu(T)$ across features in a deterministic, fully specified way. The weights $w_i(T)$ are uniquely determined by the $\mu$-coefficients and the redundancy structure; there are no user-chosen thresholds, regularization parameters, or tuning constants. For this reason, classical “hyperparameter ablations” are not meaningful here: there is nothing in ReSHAP itself to vary. We will clarify these points in the revision, emphasizing (i) that ReSHAP is parameter-free on top of existing SHAP estimators, and (ii) that its feature-size dependence and computational complexity are characterized analytically rather than controlled by hyperparameters.

---

### Official Review · Reviewer_sZKq · 2025-10-31

**Soundness:** 2
**Presentation:** 1
**Contribution:** 2
**Rating:** 2
**Confidence:** 3

**Summary:**

The paper raises a problem that when duplicate features exist, Shapley-value based explanations may be distorted. For example, the contribution of the remaining features can be significantly diminished. To address this problem, the paper proposes two properties equal division and duplication-invariance. The paper then proposes Reshap that satisfies the duplication invariant property.

**Strengths:**

1. The problem raised about duplicate or similar features distorting Shapley-value based explanations is valid and significant.

**Weaknesses:**

1. The paper is not well-written and lacking clear definitions, justifications and intuitions. This makes it hard to understand and assess the significance or soundness of the claims.
    * Why does the paper propose Lemma 3 and 2? Why do we need this alternative form of Shapley value?
    * The use and significance of Harsanyi dividends should be explained.
    * What is the formal definition of equal division in Observation 4? Equal division property does not seem to be proposed by the reference van den Brink & Funaki (2025). They propose an axiomatisation for the equal value solution instead.
    * It is hard to interpret Figure 1. What does each number represent? Why is there two 2s for D?
    * Justify why it is reasonable to define a redundant feature based on the _existence_ of a subset where $j$ adds no value. Why should the total attribution value stay the same when the value of other subsets may change (Definition 7)?
    * The proof of theorem 9 is also unclear. Can you give the function value for every subset? Why should the attribution of feature C remain unchanged in both instances?
    * Can Reshap handle more realistic similar features (instead of duplicates)?
2. The experiments section is severely lacking. Only one dataset and model is used. More information should be given about how the P and R (precision and recall?) are computed.
3. There are undiscussed existing work that consider replication robustness and propose solutions. See Han, D., Wooldridge, M., Rogers, A., Ohrimenko, O., & Tschiatschek, S. (2020). Replication Robust Payoff Allocation in Submodular Cooperative Games. IEEE Transactions on Artificial Intelligence, 4, 1114-1128.

**Questions:**

See weaknesses.

---

> ### Author Response · Authors · 2025-11-26
>
> We would like to thank the Reviewer for their careful evaluation of the paper and for the remarks that may help improve it in the future. Given the preliminary rating, we must realistically assess the likelihood of acceptance. Nevertheless, we would like to respond to the reviewers’ questions, out of full respect for their work and to provide the requested clarifications. We would appreciate the opportunity for discussion and feedback in order to better understand how to strengthen the work and address the current gaps.
>
> In the following, we address the Reviewer’s comments in detail:
>
> **1. The paper is not well-written and lacking clear definitions, justifications and intuitions. […]**
>
> We thank the Reviewer for the honest feedback regarding the writing quality. We would also like to note that another reviewer (PiXE) appreciated the quality of the writing. Nevertheless, we will make an extra effort in the future to further improve the quality of the writing.
>
> **Why does the paper propose Lemma 3 and 2? Why do we need this alternative form of Shapley value? […]**
>
> $\mu$ values are simply a change of basis in the vector space of set-function values. The entire approach could, in principle, be expressed directly using the standard $\nu $ values, but this would make the overall result significantly more intertwined. In particular, Lemma 3 provides a clear interpretation of how feature attributions are distributed in the presence of redundancy.
>
> **The use and significance of Harsanyi dividends should be explained.**
>
> Although the result does not use Harsanyi dividends, it does share some similarities. Both approaches can be viewed as changes of basis. However, they differ significantly. Harsanyi dividends decompose along subset-inclusion relationships, whereas ReSHAP along intersection relationships. We will provide a more detailed explanation in future revisions.
>
> **What is the formal definition of equal division in Observation 4 […]?**
>
> The equal-division property is not an axiom, it is a property that joint attributions are equally divided between involved players. It comes directly from symmetry axiom and in this sense Theorem 9 can be phrased also in the way that duplicate invariance property cannot be satisfied together with symmetry axiom.
>
> **It is hard to interpret Figure 1. What does each number represent? Why is there two 2s for D?**
>
> The values in Figure 1 (as explained in Figure’s caption) are the values of $\mu$. They can be interpreted as cardinalities of atomic subsets. In particular, $\mu(D)=4$ can be interpreted as the cardinality of set $D \setminus C$, $\mu(C)=2$ as the cardinality of set $C \setminus D$, and $\mu(D \cup C)=12$ can be interpreted as the cardinality of set $D \cap C$ in the Venn diagram.
>
> **Justify why it is reasonable to define a redundant feature […]?**
>
> If the condition in Definition 7 holds, it implies that $ \mu(\{j\}) = 0 $, since the contribution of feature j is fully overlapped by other subsets. In other words, feature j brings no additional information to the model because it is entirely contained within the features that cover j in the Venn diagram.
>
> **The proof of theorem 9 is also unclear. […]**
>
> It should remain unchanged, since both instances (after adding features A, B and D, respectively) form the same instance. Therefore, the contribution of C should be identical in both cases under duplicate invariance.
>
> **Can Reshap handle more realistic similar features (instead of duplicates)?**
>
> Definitely! The duplicate-feature paradox arises from duplicating a feature, but as Definition 7 indicates, ReSHAP can handle any instance involving redundant features, including even the cases where features are not pairwise strongly correlated yet one feature is still fully replaceable by a subset of others. Appendix B showcases such an instance in the simplest possible setting, making it relatively easy to follow.
>
> **2. The experiments section is severely lacking […]**
>
> Although it is hard to disagree with this observation, we would like to emphasize the strong fundamental contribution of our result, which solves a well-known problem of Shapley values. Moreover, the current computational complexity makes it difficult to conduct extensive experimental validation. However, this limitation can be possibly addressed in future work by following the line of research established for SHAP and translating existing approximation techniques to the ReSHAP setting (see Appendix F).
>
> **3. There are undiscussed existing work […].**
>
> This work is actually very interesting! We sincerely thank the Reviewer for pointing it out, we were not aware of it, as it is framed within a somewhat different setting. For the sake of space, we would like to highlight one key difference between the approaches: while Han et al. require submodularity of $ \nu $, our method operates without this assumption. Nevertheless, we would be very happy to include this result in our discussion in the paper.

---

### Official Review · Reviewer_YN5V · 2025-10-31

**Soundness:** 3
**Presentation:** 2
**Contribution:** 1
**Rating:** 2
**Confidence:** 4

**Summary:**

I have just reviewed this manuscript for the AAAI.  As far as I can tell, the manuscript has been reformatted for the ICLR and added a page of 'Analysis and Empirical Evaluation' ($\S 7$), but not otherwise to have taken into account my previous review.

That review can be found at https://openreview.net/forum?id=p8Zao3l8HP&noteId=viyMisQReq

> The Shapley value feature attribution technique randomises uniformly over features when seeking to assign 'value-added' to a feature. Thus, duplicating a feature gives that feature twice as many opportunities to add value - even though no explanatory power has been added to the model. (See Kumar et al. 2020 - as cited by the authors - for an example.)

> The paper introduces ReSHAP, which seeks to correct for interaction effects between features.

> It does so by decomposing the Shapley value using Möbius transforms, following a tradition dating back at least to Grabisch & Roubens (1999); see Pymar et al. (2022) for a comparison of some of the 'interaction' literature, and Stoian (2023) for a recent paper on using this decomposition for faster computations.

> The authors, however, compare their approach to 'correlation-based' approaches - giving the impression that they are not familiar with the existing literature.

> Thus, at this point, it is not clear what distinctly novel issues are introduced by this paper:

> 1. the motivating concern is known.
> 1. no comparison with existing Shapley/Möbius/interaction methods is carried out.
> 1. it does not re-analyse the existing literature in a way that clarifies unresolved issues. (Indeed, it does not demonstrate a tight grasp on it: e.g. the claim that "computation is not guaranteed in general" is unclear in Frye et al., which uses a deterministic algorithm with a guaranteed, unique solution, given some encoding of domain knowledge.)

> I would encourage the authors to focus their rebuttal on how ReSHAP compares to existing techniques.

This remains my view.

**Strengths:**

See above: understanding how features interact in ML models is an important problem.

**Weaknesses:**

See above: although this problem has already been studied, the authors do not compare their approach to the most similar approaches in the literature.

When I first reviewed this paper for AAAI, I wondered whether they were just not familiar with the literature.  Having cited papers in the literature, and recommended that the authors compare their approach to existing ones, that interpretation is no longer tenable.

**Questions:**

See above.

---

> ### Author Response · Authors · 2025-11-26
>
> We would like to thank the Reviewer for their careful evaluation of the paper and for the remarks that may help improve it in the future. Given the preliminary rating, we must realistically assess the likelihood of acceptance. Nevertheless, we would like to respond to the reviewers’ questions, out of full respect for their work and to provide the requested clarifications. We would appreciate the opportunity for discussion and feedback in order to better understand how to strengthen the work and address the current gaps.
>
> First of all, we would like to thank the Reviewer for the valuable feedback provided earlier during the AAAI review process, which already helped us improve the manuscript. In particular, we added a computational complexity analysis and practical considerations (Appendix F), a comparison with existing approaches (Appendix G), two simple synthetic examples that illustrate the construction and behavior of ReSHAP (Section 7.1 and Appendix B), and one small-scale real dataset experiment (Section 7.1 and Appendix E).
>
> In the following, we address the Reviewer’s comments in detail:
>
>
> **It does so by decomposing the Shapley value using Möbius transforms, following a tradition dating back at least to Grabisch & Roubens (1999); see Pymar et al. (2022) for a comparison of some of the 'interaction' literature, and Stoian (2023).**
>
> We would like to thank the Reviewer for pointing out these three papers. We are aware of these results. The literature on Shapley values and their practical variations is very broad. Below, we briefly explain why these three works were not selected among the most central references for our contribution.
>
> The work by **Grabisch & Roubens (1999)** defines an interaction index for each subset \(S\). Importantly, their construction does **not** resolve the duplicate-feature paradox or the redundancy problem, precisely because it satisfies the symmetry axiom. Its main purpose is to measure how players interact, conceptually a very different direction. What the two approaches share is the use of a Möbius-type transformation, to compute a new basis in the vector space of set-function values. However, these bases serve fundamentally different purposes: Grabisch & Roubens use a **subset-based** Möbius transform to measure interactions, whereas ReSHAP relies on an **intersection-based** Möbius transform to measure redundancy.
>
> The second reference, **Pymar et al. (2022)**, also goes in a different direction. While Grabisch & Roubens aim to understand internal interactions within a subset $S$, Pymar et al. focus on joint interactions among features. Importantly, their result does not aim to address the redundancy problem among features, and for this reason, we did not consider it among the core references for our work.
>
> Regarding the third reference, in **Stoian (2023)**, the contribution is centered on reducing the computational complexity of the Joint Shapley Values. Since JSV do not aim to solve the redundancy problem, this line of work does not directly align with our goals. However, Stoian’s results may become highly relevant for future efforts to reduce the computational complexity of ReSHAP, one of our main directions for follow-up work.
>
> Finally, we would like to point out that, among the works that address problems closely related to ours, Frye et al. (2020), Kwon & Zou (2022), Watson et al. (2023), and Ay et al. (2020), **none** cite **any** of the three references mentioned above.
>
> To conclude, we greatly appreciate these references; however, we did not include them in the bibliography because we found other results to be more directly connected to the core contributions of our work. That said, we are open to add them in future revisions, in the context of mathematical techniques and computational complexity reductions for ReSHAP.
>
> **The authors, however, compare their approach to 'correlation-based' approaches - giving the impression that they are not familiar with the existing literature.**
>
> We thank the Reviewer for the valuable feedback. In the current version in this section, we focus primarily on showing that correlation-based approaches cannot solve the duplicate-feature paradox or the redundancy problem, not on the comparison. We have also moved this discussion from the main body to Appendix C.
>
> **It does not re-analyse the existing literature in a way that clarifies unresolved issues. (Indeed, it does not demonstrate a tight grasp on it: e.g. the claim that "computation is not guaranteed in general" is unclear in Frye et al., )**
>
> We thank the Reviewer for valuable input. The narrative has been substantially improved, including the imprecise statements that the Reviewer correctly pointed out. In particular, the sentence highlighted by the Reviewer was revised to: “such as Frye et al. (2020), reweight permutations to improve attribution, yet lack a universal principle for selecting weights.”

---

> > ### Comment · Reviewer_YN5V · 2025-11-28
> > **look forward to seeing a proper re-drafting**
> >
> > It remains unclear to me why a literature (dating back to Grabisch & Roubens) that is designed to address non-independence of features (under whatever name) does not - in the authors' view.
> >
> > Thus, it remains my view that this should be properly addressed in the paper itself, rather in the rebuttal period - in which one is more limited (e.g. to assertions).  I would therefore welcome the chance to read a proper re-drafting of the paper that properly engaged with my questions.

---

> > > ### Author Response · Authors · 2025-12-04
> > >
> > > As the discussion period comes to an end, we would like to thank the reviewer for their work and the valuable discussion. Following the reviewer’s comments, in the next iteration we will more carefully address the points raised, in particular the proposed literature.

---

### Official Review · Reviewer_PiXE · 2025-10-31

**Soundness:** 2
**Presentation:** 3
**Contribution:** 2
**Rating:** 4
**Confidence:** 5

**Summary:**

This paper introduces ReSHAP, a redundancy-weighted generalization of Shapley-based feature attribution. The method addresses the duplicate-feature paradox - the well-known issue that adding redundant or duplicate features changes attribution results in standard SHAP. The authors propose a recursive redistribution algorithm (Algorithm 1) to adjust the Möbius-based Shapley decomposition and theoretically prove that ReSHAP satisfies duplication invariance. Experiments on a small regression dataset illustrate improved stability of feature importance. The conceptual motivation of ReSHAP is strong and theoretically well presented. However, the algorithm currently lacks comprehensive experimental validation and requires deeper discussion of computational feasibility.

**Strengths:**

S1) The paper is clearly written and well structured, making the theoretical content relatively accessible.
S2) Theorem 9 provides a meaningful theoretical insight into the trade-off between equal division and duplication invariance.
S3) The recursive redistribution algorithm is original and intuitively motivated.

**Weaknesses:**

W1) The weighting scheme in ReSHAP is defined heuristically. (D1)
W2) The experimental evidence can be improved. (D2, D4)
W3) Key results such as complexity analysis and comparisons with baselines are placed in the appendix, which weakens their visibility. (D2, D3, D4)
W4) The algorithmic complexity is prohibitively high; the paper should discuss the feasibility of practical computation.

**Questions:**

D1) Although Theorem 11 justifies duplication invariance, the derivation of the weighting rule lacks rigorous theoretical grounding. It would strengthen the paper to provide an analytical motivation for the weighting rule.
D2) The experiment uses only the Ames Housing dataset and a simple MLP model. Additional datasets of varying dimensionality and correlation structure would better demonstrate generality.
D3) Appendix E indicates that the number of features is only about a dozen. A discussion on how ReSHAP scales to higher-dimensional data, or an approximation scheme for large n, would be valuable.
D4) Appendix G contains no experimental comparison with SHAP or related redundancy-aware methods. Including such baselines—or at least a discussion of expected differences—would make the empirical section more convincing.

---

> ### Author Response · Authors · 2025-11-26
>
> We would like to thank the Reviewer for their careful evaluation of the paper and for the remarks that may help improve it in the future. Given the preliminary rating, we must realistically assess the likelihood of acceptance. Nevertheless, we would like to respond to the reviewers’ questions, out of full respect for their work and to provide the requested clarifications. We would appreciate the opportunity for discussion and feedback in order to better understand how to strengthen the work and address the current gaps.
>
>
> **The weighting scheme in ReSHAP is defined heuristically.**
>
> We thank the Reviewer for a very valid point. A more rigorous technical grounding would indeed strengthen the paper.
> There is, however, rigorous technical grounding: under the information-redundancy measure we use, ReSHAP is the unique method that resolves both the duplicate-feature paradox. This follows from the decomposition of attributions along the intersection relationships encoded in the $\mu$ function. It is also worth noting that other notions of information redundancy (e.g., those based on KL divergence) would naturally lead to different weighting rules. We would be glad to add this to the body of the paper.
>
> **The experimental evidence can be improved.**
>
> The experimental evidence is based on one real-world dataset and two simple synthetic examples. We agree with the Reviewer that the experimental section is rather short. However, as the Reviewer correctly pointed out, the current computational complexity makes it difficult to conduct extensive experimental validation.
>
> Although a deeper empirical investigation would certainly strengthen the experimental support, we emphasize that the proposed ReSHAP method is theoretically proven to resolve the duplicate-feature paradox and to correctly attribute redundant features (Theorem 11). From this perspective, the experiments primarily illustrate the behavior of ReSHAP on real-world data rather than serve as empirical confirmation of the method’s correctness.
>
> **Key results such as complexity analysis and comparisons with baselines are placed in the appendix, which weakens their visibility.**
>
> We thank the Reviewer for pointing this out, and we agree that placing it in the main body would increase the visibility of these parts. We will definitely move them to the main body.
>
> **The algorithmic complexity is prohibitively high; the paper should discuss the feasibility of practical computation, [...] approximation scheme for large n, would be valuable.**
>
> We fully agree with the reviewer that the computational complexity is high. As presented in Appendix F, it is exponential in n. However, it is important to point out that this work aims to investigate the fundamental limitations of Shapley values, which, in their pure form, also require exponential computations in n. Thus, a foundational improvement to the Shapley value framework is a necessary step toward resolving its inherent limitations and an exponential running time is expected therein.
>
> That said, it is worth emphasizing that the machinery presented in this paper is not purely theoretical. A key component of our approach is that it relies solely on evaluations of the set function $\nu$. As discussed in Appendix F, methods such as PermutationSHAP, KernelSHAP, may potentially be adapted to yield efficient procedures capable of resolving the duplicate-feature paradox. Due to the depth of this research direction, we leave these developments for future work.
>
> **Appendix G contains no experimental comparison with SHAP or related redundancy-aware methods.**
>
> We thank the Reviewer for the comment. Clearly, expanding the experimental section to include a comparison with other methods would further strengthen the paper. However, among existing methods, only two approaches are potentially capable of addressing the duplicate-feature paradox and redundancy problem: Frye et al. (2020) and Kwon & Zou (2022). Other methods, such as Watson et al. (2023) and Ay et al. (2020), do not break the symmetry axiom and, according to Theorem 9, therefore cannot resolve the issue.
>
> Experimental comparison with Frye et al. (2020) is challenging because, although their method aims to modify weights (and thus violate symmetry), it focuses on axiomatic properties and does not provide an agnostic, generally applicable procedure for doing so. The weightening procedure is not clearly defined in general. Comparison with Kwon & Zou (2020) is possible, though their weight-learning scheme requires training an auxiliary ML model to estimate data-value weights. Unfortunately, this introduces multiple layers of complexity, both in computing the $\nu$ values and in the weighting mechanism, which may obscure the comparison and make a direct evaluation difficult. Finally, it is worth pointing out that neither of these methods is designed to solve the redundancy problem, ReSHAP is the first method that provably does so.

---

### Author Response · Authors · 2025-12-04

As the discussion period comes to an end, we would like to thank all the Reviewers for their work and the valuable discussion. We will carefully consider all comments and address the relevant points in the next iteration of the manuscript.

---

### Note · Authors · 2025-12-26

**Comment:**

Based on the constructive feedback received and the identified directions for improvement, we have chosen to withdraw the paper from ICLR and revise it prior to submission to another venue.

**Withdrawal Confirmation:**

I have read and agree with the venue's withdrawal policy on behalf of myself and my co-authors.